# Comparative outcomes of combined corticosteroid and remdesivir therapy with corticosteroid monotherapy in ventilated COVID-19 patients

**Subhadra Mandadi**[1]*, **Harish Pulluru**[2]°, **Frank Annie**[3]°

**1** Department of Infectious Diseases, Charleston Area Medical Center, Charleston, West Virginia, United States of America, **2** Department of Hospital Medicine, Charleston Area Medical Center, Charleston, West Virginia, United States of America, **3** Charleston Area Medical Center Health Education and Research Institute, Charleston, West Virginia, United States of America

° These authors contributed equally to this work.

* Subhadra.mandadi@gmail.com

**Data Availability Statement:** The legal and ethical restrictions under which the data were provided do not allow for the data to be made publicly available. The data we used for this paper were acquired

## Abstract

Remdesivir (RDV) reduces time to clinical improvement in hospitalized COVID-19 patients requiring supplemental oxygen. Dexamethasone improves survival in those requiring oxygen support. Data is lacking on the efficacy of combination therapy in patients on mechanical ventilation. We analyzed for comparative outcomes between Corticosteroid (CS) therapy with combined Corticosteroid and Remdesivir (CS-RDV) therapy. We conducted an observational cohort study of patients aged 18 to 90 with COVID-19 requiring ventilatory support using TriNetX (COVID-19 Research Network) between January 20, 2020, and February 9, 2021. We compared patients who received at least 48 hours of CS-RDV combination therapy to CS monotherapy. The primary outcome was 28-day all-cause mortality rates in propensity-matched (PSM) cohorts. Secondary outcomes were Length of Stay (LOS), Secondary Bacterial Infections (SBI), and MRSA (*Methicillin-Resistant Staphylococcus aureus*), and *Pseudomonas* infections. We used univariate and multivariate Cox proportional hazards models and stratified log-rank tests. Of 388 patients included, 91 (23.5%) received CS-RDV therapy, and 297 (76.5%) received CS monotherapy. After propensity score matching, with 74 patients in each cohort, all-cause mortality was 36.4% and 29.7% in the CS-RDV and CS therapy, respectively (P = 0.38). We used a Kaplan-Meier with a log-rank test on follow up period (P = 0.23), and a Hazards Ratio model (P = 0.26). SBI incidence was higher in the CS group (13.5% vs. 35.1%, P = 0.02) with a similar LOS (13.4 days vs. 13.4 days, P = 1.00) and similar incidence of MRSA/*Pseudomonas* infections (13.5% vs. 13.5%, P = 1.00) in both the groups. Therefore, CS-RDV therapy is non-inferior to CS therapy in reducing 28-day all-cause in-hospital mortality but associated with a significant decrease in the incidence of SBI in critically ill COVID-19 patients.

from TriNetX (https://www.trinetx.com/). Release and/or sharing of these data are not covered under our data use agreement with TriNetX. To gain access to the data, a request can be made to TriNetX (moc.xtenirt@nioj), but costs may be incurred, and a data sharing agreement would be necessary.

**Funding:** The author(s) received no specific funding for this work.

**Competing interests:** The authors have declared that no competing interests exist.

## Introduction

COVID-19 dominated 2020, emerging as a global pandemic, created havoc since its emergence as a zoonotic disease in China, 2019, causing death surge and economic devastation. Several therapeutic agents have been evaluated for the treatment of COVID-19. However, no antiviral agents were shown to be effective, especially in COVID-19 illness requiring ventilatory support associated with high mortality rates. RDV, a repurposed antiviral agent, is currently the only drug approved by the Food and Drug Administration (FDA) to treat COVID-19 hospitalized patients who require supplemental oxygen [1]. The results were primarily based on the multinational, double-blind, randomized controlled trial that showed a reduction in clinical recovery time with RDV use in hospitalized patients with severe disease [2]. Dexamethasone, a glucocorticoid, has been found to improve survival in hospitalized patients who require any oxygen support [3], and its use was strongly recommended.

There is a lack of consensus on RDV use in patients requiring ventilatory support. WHO (World Health Organization) recommends against RDV use outside of clinical trials for COVID-19 of any disease spectrum [4]. IDSA (Infectious Disease Society of America) COVID-19 treatment guidelines suggest against the routine use of RDV in patients requiring ventilatory support [5], while NIH (National Institute of Health) treatment guidelines suggest considering its use in combination with dexamethasone [6]. These circumstances have enabled independent institutional policies regarding RDV use with no clear guidance.

## Materials and methods

### Ethics statement

Our study was approved under exemption by the CAMC (Charleston Area Medical Center) research and Grant's administration Institutional Review Board (study number 21–723) and received a waiver of informed consent. The study used data from TriNetX, a global federated health research network that provided an anonymized dataset of electronic medical records (EMRs). TriNetX is compliant with the Health Insurance Portability and Accountability Act (HIPAA), the US federal law that protects healthcare data privacy and security. TriNetX is certified to the ISO 27001:2013 standard and maintains an Information Security Management System (ISMS) to ensure the protection of the healthcare data it has access to and meet the HIPAA Security Rule requirements. Any data displayed on the TriNetX Platform in aggregate form, or any patient level data provided in a data set generated by the TriNetX Platform, only contains de-identified data as per the de-identification standard defined in Section §164.514(a) of the HIPAA Privacy Rule. The process by which the data is de-identified is attested to through a formal determination by a qualified expert as defined in Section §164.514(b)(1) of the HIPAA Privacy Rule. This formal determination by a qualified expert, refreshed in December 2020, supersedes the need for TriNetX's previous waiver from the Western Institutional Review Board (IRB). The TriNetX network contains data provided by participating Healthcare Organizations (HCOs), each of which represents and warrants that it has all necessary rights, consents, approvals, and authority to provide the data to TriNetX under a Business Associate Agreement (BAA), so long as their name remains anonymous as a data source and their data are utilized for research purposes. The data shared through the TriNetX Platform are attenuated to ensure that they do not include sufficient information to facilitate the determination of which HCO contributed specific information about a patient. Further details about TriNetX processes and standardization of data are provided in S1 Text.

## Study design and data source

The design is a cohort study comparing patients with critical COVID-19 illness treated with CS-RDV combination therapy versus CS monotherapy. Using the TriNetX network, a deidentified dataset of COVID-19 patients with a PCR confirmed SARS-COV-2 diagnosis, admitted to the Intensive Care Unit (ICU) aged 18 to 90, was identified in EMRs between January 20th, 2020, and February 9th, 2020. For this study, we accessed the data from health care organizations in the United States.

## Study population

We queried the COVID-19 research network, a collection of 60 health care organizations, from January 20th, 2020, to February 9th, 2021. All the patients who were 18–90 years of age with PCR confirmed SARS-CoV-2 test admitted requiring ventilatory support were identified using proper diagnostic codes (Table 1). Additional inclusion criteria for study arms of CS-RDV and CS therapy were the presence of radiographic evidence of pulmonary infiltrates and the use of therapies for at least 48 hours of hospitalization. Exclusion criteria for both study arms were pregnant or breast-feeding women and known allergic reactions to the treatments mentioned. The patients who expired within 48 hours of hospitalization were excluded. While retrospectively selecting patients for the CS-RDV group from the database, we did not include patients with elevated transaminases more than five times the standard normal upper limit. Patients who were included received RDV intravenous (IV) as a 200 mg loading dose, followed by a 100 mg maintenance dose on days 2–5 or until hospital discharge or death. Patients received dexamethasone 6 mg daily, IV or equivalent doses of methylprednisolone 32mg daily, or hydrocortisone 160mg daily. For the combination therapy study arm, patients sequentially received CS therapy, followed by RDV on day 1 of admission.

## Outcome measures

The primary outcome was 28-day all-cause in hospital mortality rates after 48 hours of therapy and five days of therapy. The secondary outcome measures were LOS, SBI, and infections with MRSA (*Methicillin Resistant Staphylococcus aureus*) and *Pseudomonas* species.

**Table 1. Diagnostic codes.**

| Code–ICD 10 | Description |
| --- | --- |
| **SARS–COV– 2 Lab Codes** | |
| B34.2 | Coronavirus Infection |
| B97.29 | Other Coronavirus |
| J12.01 | Pneumonia due to SARS-associated coronavirus |
| U07.1 | 2019-NCOV acute respiratory disease |
| 94307–6 | SARS coronavirus 2 N gene (Presence) |
| 94308–4 | SARS coronavirus |
| 94310–0 | SARS-like Coronavirus N gene (Presence) |
| 94314–2 | SARS coronavirus 2 RdRp gene (Presence) |
| 94315–9 | SARS coronavirus 2 E gene (Presence) |
| 94316–7 | SARS coronavirus 2 N gene (Presence) |
| **Corticosteroids** | |
| 5492 | Hydrocortisone |
| 6902 | Methylprednisolone |
| 3264 | Dexamethasone |
| **Remdesivir** | |
| 2284718 | Remdesivir |

## Data analysis

To measure potential differences of the constructed cohorts, we used descriptive statistics like the mean ± standard deviation for continuous measures. To further explore differences, we used a chi-square test for categorical variables. We used the TriNetx online platform to match the different cohorts with a 1:1 propensity match using logistic regression to create two well-matched groups. The TriNetx platform uses logistic regression to obtain listed propensity scores for each of the selected literature-driven covariates. The Propensity score matching (PSM) utilizes the Python libraries (Numpy and sklearn). The PSM platform also runs the results in R to compare and verify the models and output. A final step of verification uses a nearest neighbor function set to a tolerance level of 0.01 and a difference value of $> 0.1$. All-cause mortality for the PSM was determined using a Kaplan-Meier and log-rank test with a 28-day period.

To understand if differing health outcomes affected the conditions driving all-cause mortality, we conducted two sensitivity analyses. Given the possibility of residual confounders, we used the falsification endpoint of bleeding, which would likely not be affected by SARS-COV-2 and the treatment plan examined within this study. We also created two similar cohorts with differing time frames from the main study, which did not include the 48 hours, to verify the results.

## Results

### Patient characteristics

Of 388 critically ill COVID-19 patients requiring ventilatory support, 91 cases (23.5%) who received CS-RDV therapy for at least 48 hours were included in the first cohort. The second cohort included individuals who received CS therapy for at least 48 hours, totaling 297 cases (76.5%). As shown in Table 2, our study noted no differences in age and sex distribution between the cohorts. We had an exceptionally low sample size in matched cohorts belonging to the Asian race making its P value significant. In terms of preexisting chronic conditions, patients who received CS therapy had a higher prevalence of hypertension (75% vs. 88%; P = 0.03), diabetes (53% vs. 67%; P<0.01), congestive heart failure (37% vs. 55%; P = 0.04), coronary artery disease (34% vs. 46%; P = 0.04), chronic kidney disease (33% vs. 50%; P<0.01), chronic obstructive pulmonary disease (27% vs. 39%; P = 0.04), and a lower prevalence of chronic liver disease (19% vs. 28%; P = 0.04) than patients who received CS-RDV therapy among unmatched cohorts. Patients in the unmatched CS-RDV group had higher use of convalescent plasma (20% vs. 3%, P<0.01) than those who received CS.

### Outcome measures

After propensity score matching (74/74), 28-day all-cause mortality was similar in the CS-RDV and CS groups (36.5% vs. 29.7%, P = 0.38) after 48 hours of therapy (Table 3). A log rank-test also confirmed no difference in mortality at the end of the survival probability of 28 days (58% vs. 66%, P = 0.23) (Fig 1). A hazard ratio confirmed no difference in the matched cohorts (P = 0.26). The length of stay was similar between the CS-RDV and CS groups (13.4 days vs. 13.4 days, P = 1.00) (Table 3). SBI incidence was higher in the CS group (35.1% vs 13.5%, P = 0.02) with a similar incidence of MRSA/*Pseudomonas* infections (13.5 vs. 13.5, P = 1.00) (Table 3). A log rank test also confirmed the difference in incidence of SBI at the end of the survival probability of 28 days (94.5% vs. 59.9%, P < 0.01) (Fig 2) but was not shown to be an independent predictor of mortality (37.5% vs 85.7%, P = 0.10) (Fig 3).

**Table 2. Baseline characteristics.**

| Baseline Characteristics | Unmatched Cohorts | | | | Propensity Matched Cohorts | | | |
|---|---|---|---|---|---|---|---|---|
| | CS-RDV (91) | CS (297) | P-Value | SMD | CS-RDV (74) | CS (74) | P-Value | SMD |
| Age | 61.7±14.7 | 61.5± 14.6 | 0.92 | 0.01 | 61.9± 15.2 | 61.3±14.9 | 0.79 | 0.04 |
| Male | 56% | 53% | 0.63 | 0.06 | 57% | 45% | 0.14 | 0.25 |
| Female | 44% | 47% | 0.63 | 0.06 | 43% | 55% | 0.14 | 0.25 |
| White | 58% | 57% | 0.78 | 0.03 | 58% | 64% | 0.50 | 0.11 |
| Black or African American | 36% | 31% | 0.35 | 0.11 | 35% | 31% | 0.60 | 0.09 |
| Hispanic or Latino | 11% | 7% | 0.23 | 0.14 | 14% | 14% | 1.00 | 0.01 |
| Asian | 11% | 3% | 0.04 | 0.30 | 14% | 0% | 0.01 | 0.56 |
| Hypertension | 75% | 88% | 0.03 | 0.33 | 77% | 78% | 0.84 | 0.03 |
| Diabetes | 53% | 67% | <0.01 | 0.30 | 53% | 51% | 0.87 | 0.03 |
| Congestive Heart Failure | 37% | 55% | 0.04 | 0.35 | 43% | 46% | 0.74 | 0.05 |
| Chronic Kidney Disease | 33% | 50% | <0.01 | 0.34 | 38% | 38% | 1.00 | 0.01 |
| Coronary Artery Disease | 34% | 46% | 0.04 | 0.25 | 36% | 38% | 0.86 | 0.03 |
| Chronic Obstructive pulmonary disease | 27% | 39% | 0.04 | 0.25 | 30% | 28% | 0.86 | 0.03 |
| History of Stroke | 20% | 23% | 0.53 | 0.08 | 19% | 24% | 0.42 | 0.13 |
| Smoking History | 16% | 22% | 0.27 | 0.14 | 16% | 19% | 0.67 | 0.07 |
| Transplantation | 15% | 11% | 0.27 | 0.13 | 15% | 15% | 1.00 | 0.01 |
| chronic liver disease | 19% | 28% | 0.04 | 0.20 | 22% | 23% | 0.90 | 0.02 |
| Obesity (BMI>/ = 30) | 73% | 66% | 0.24 | 0.14 | 69% | 73% | 0.59 | 0.09 |
| Convalescent Plasma | 20% | 3% | <0.01 | 0.53 | 14% | 14% | 1.00 | 0.01 |

BMI, Body Mass Index; CS-RDV, Corticosteroid-Remdesivir; CS, Corticosteroid; SMD, Standard Mean Difference.

A Sensitivity Analyses was performed using a falsification endpoint of bleeding in the 48-hour period. No difference in the falsification endpoint was observed with a log-rank test (P = 0.23), suggesting the absence of possible unmeasured confounders that affected the explored outcomes in this study. To confirm that no data were censored during the 48-hour period, we conducted a sub-analysis and removed the time variable of at least 48 hours and reexamined the data cohorts. We identified a total of 461 patients in this sub-analysis. Of those patients, 121 cases (26.2%) who received CS-RDV therapy were identified as the first cohort. The second cohort that received CS monotherapy totaled 340 cases (73.8%). After propensity score matching (100/100) of the same original covariates, all-cause mortality was similar in the two cohorts (31% vs. 26%, P = 0.43). A log rank-test confirmed no difference in mortality at the end of the survival probability of 28 days (63.7% vs. 70.5%, P = 0.47). Overall, it appeared that there was no difference in all-cause mortality and LOS in the compared cohorts.

**Table 3. Outcome measures.**

| Outcome measures | Unmatched Cohorts | | | Propensity Matched Cohorts | | |
|---|---|---|---|---|---|---|
| | CS-RDV (91) | CS (297) | P-Value | CS-RDV (74) | CS (74) | P-Value |
| All Cause Morality | 33% | 23% | 0.10 | 36.5% | 29.7% | 0.38 |
| Length of Stay (days) | 14.4 | 15 | 0.13 | 13.4 | 13.4 | 1.00 |
| SBI | 11% | 27% | 0.03 | 13.5% | 35.1% | 0.02 |
| MRSA/*Pseudomonas* infections | 11% | 3.4% | 0.04 | 13.5% | 13.5% | 1.00 |

CS-RDV, Corticosteroid-Remdesivir; CS, Corticosteroid; MRSA, *Methicillin Resistant Staphylococcus Aureus*; SBI, Secondary Bacterial Infections.

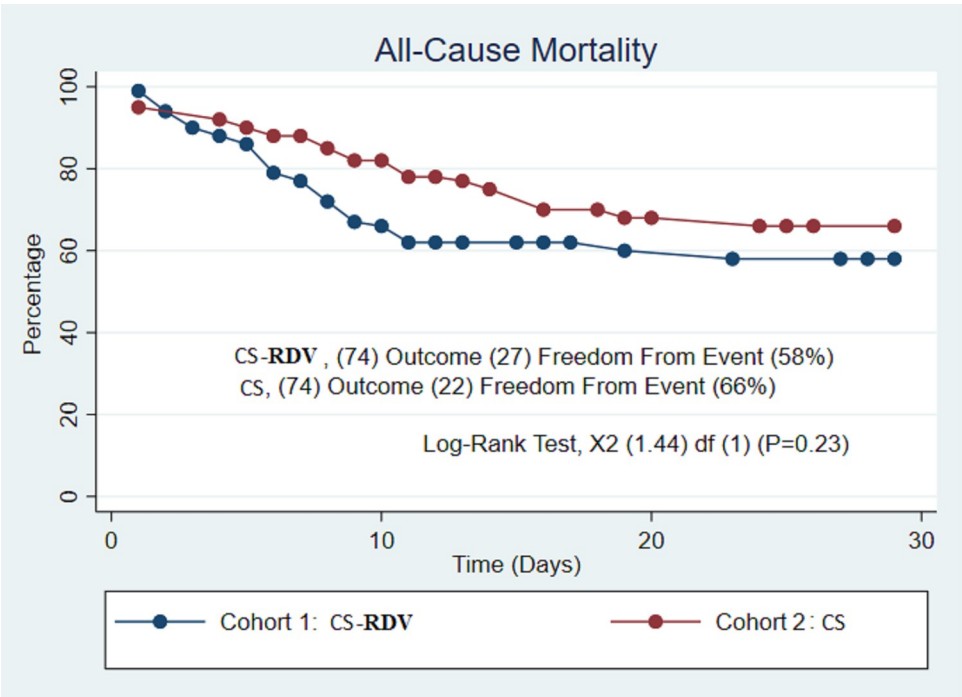

**Fig 1. All-cause mortality rates.** Kaplan-Meier survival analysis of the study groups after propensity score matching showing no significant difference in all-cause mortality rates between CR-RDV(Corticosteroid-Remdesivir) therapy arm and CS (Corticosteroid) therapy arm.

With the mortality rates obtained, a 2-sided test with 80% power and a P value of 0.05 determined a minimum sample size of 140 (70 in each arm) would be required to detect the difference between the two groups.

## Discussion

Our study reported that CS-RDV therapy's comparative outcomes with CS therapy in patients with PCR confirmed SARS-CoV 2 diagnosis and required invasive mechanical ventilation (IMV). After matching the two cohorts, all-cause 28-day mortality rates for 48 hours were similar (36.5% vs. 29.7%, P = 0.38) between CS-RDV and CS therapy, respectively. The all-cause 28-day mortality rates between matched cohorts (28/28) calculated after five days of CS-RDV and CS therapy were 51% and 70%, respectively (P = 0.11). The difference between the matched cohorts appears to be numerically significant at 19% but did not reach statistical significance, which needs further evaluation with a larger sample size. The potential confounders were adjusted, and we used falsification endpoints such as bleeding to further validate the findings. None of the patients were discharged to hospice. Thus, no difference in outcomes between the cohorts was observed. We found no difference in length of stay in both matched cohorts (13.4 days vs. 13.4 days, P = 1.00).

Mortality from COVID-19 is exceptionally high among patients with comorbidities and those who require invasive mechanical ventilation [7]. Data from a large, multicenter, randomized, open-label trial showed that dexamethasone at a dose of 6 mg daily for up to 10 days reduced 28-day mortality in patients with COVID-19 who require respiratory support (29.3% in dexamethasone group compared to 41.4% in the usual care group) [3]. The data from the prospective meta-analysis from the WHO Rapid Evidence Appraisal for COVID-19 Therapies

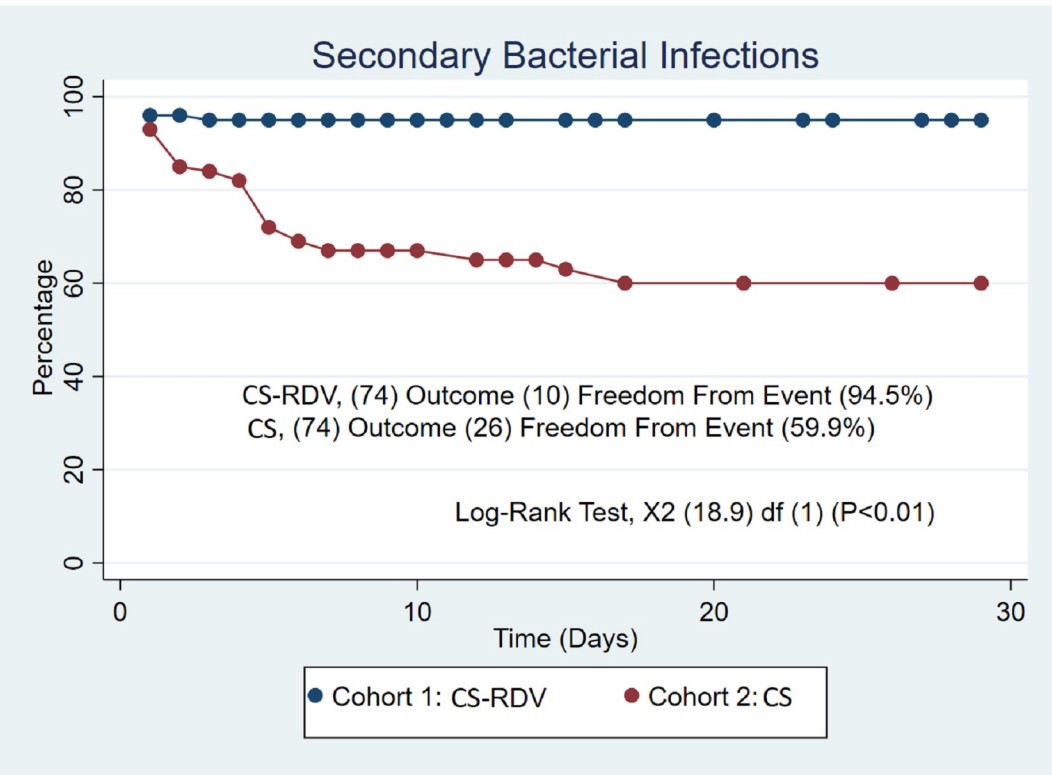

**Fig 2. Secondary bacterial infections.** Kaplan-Meier survival analysis of the study groups after propensity score matching showing higher incidence of SBI (Secondary Bacterial Infections) in CS (Corticosteroid) therapy arm compared to CS-RDV (Corticosteroid-Remdesivir) therapy arm.

(REACT) Working Group pooled data from 7 trials (RECOVERY, REMAP-CAP, CoDEX, CAPE COVID, and three additional trials), of which 59% were from the RECOVERY trial, 28-day all-cause mortality was lower among patients who received CS, further supporting the use in critically ill COVID-19 patients who require respiratory support [8]. Data from a randomized controlled trial in patients with severe COVID-19, RDV reduced clinical recovery time in hospitalized patients who required supplemental oxygen with no observed benefit in those who were on high-flow oxygen, noninvasive ventilation, mechanical ventilation, or ECMO, but the study was not powered for mortality [2]. Conversely, the SOLIDARITY trial, a multinational trial [4], showed no mortality benefit using RDV. A systematic review and meta-analysis on the efficacy and harms of RDV use in hospitalized patients with COVID-19 were considered inconclusive due to the lack of adequately powered and fully reported randomized controlled trials [9].

It is unconfirmed whether the current evidence on lack of recovery and mortality benefit in ventilated patients with RDV can be improved with concomitant CS use. There are theoretical reasons that combination therapy may be beneficial in some patients with severe COVID-19. However, the safety and efficacy have not been rigorously studied in clinical trials, especially in ventilated patients. The CS-RDV combination is being used clinically in a few institutions for severe COVID-19, given improved clinical recovery time. Our study is one step towards such understanding. Our study has shown no mortality benefit than those who received CS alone. Further studies are essential to confirm our findings.

Among the unmatched cohorts, there were significant differences in preexisting health conditions. An independent additional sensitivity analysis with Diabetes and Chronic kidney

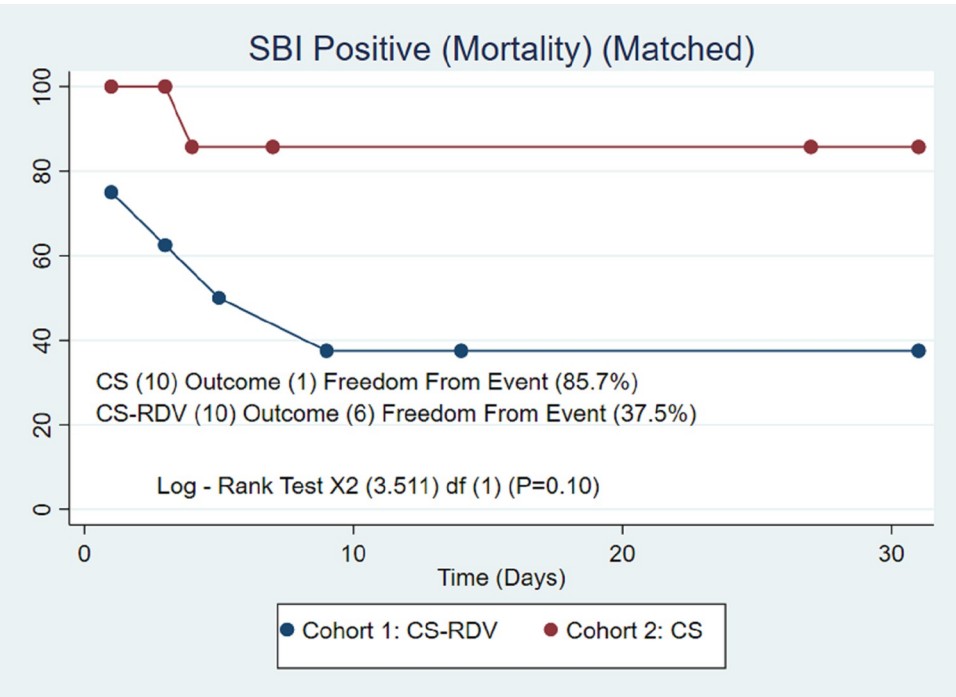

**Fig 3. Secondary bacterial infections.** Kaplan-Meier survival analysis of the study groups after propensity score matching showing no impact of SBI (Secondary Bacterial Infections) in CS (Corticosteroid) therapy arm and CS-RDV (Corticosteroid-Remdesivir) therapy arm.

disease showed no statistical measure influencing mortality rates independent of drug exposure. The results indirectly inform that the associated comorbidities might not have predictive effect on mortality rates in critically ill COVID-19 patients. An article published on autopsy findings of the 26 cases of hospitalized patients with COVID-19 evaluated the contribution of the preexisting health conditions to the risk of death. The investigators suggested that most patients whose median age was 70 years have died of COVID-19 illness with only contributory implications of preexisting health conditions to the mechanism of death [10].

Our study evaluated the incidence and impact of SBI in the two cohorts. Patients in the Intensive Care Unit (ICU) are vulnerable to SBI for a multitude of reasons including steroid use. In a living systematic review and meta-analysis on bacterial co-infection and superinfection in COVID-19 patients, 14.3% of patients had SBI, more common in critically ill patients at 8.1%, with the majority receiving antibiotics at 71.9% [11]. Data from a meta-analysis showed that 43.7% to 100% of patients received antibiotics and that 4.8% to 19.5% developed SBI and was associated with a severe course or fatal outcome [12]. We wished to know whether adding RDV in critically ill COVID-19 patients would affect the likelihood of SBI and impact the mortality. We analyzed data on blood cultures, respiratory cultures (sputum, bronchoalveolar fluid), Pneumonia PCR panel, legionella, and streptococcus pneumoniae urinary antigens after 48 hours of admission to identify patients with SBI. Among the adjusted cohorts, 35.1% of patients in our study's CS group had SBI, unlike 13.5% in patients who received combination therapy (P = 0.02) with a statistically significant difference noted in bacteremia occurrence and with no difference in the likelihood of pneumonia. Nevertheless, SBI was not shown to be an independent risk factor of mortality in our study. Most of the CS therapy cohort received a combination of steroids, and therefore, we could not determine the association of SBI with an independent steroid regimen.

A recent study done on machine learning as a precision medicine approach to identify a group of general inpatient COVID-19 patients who might benefit from COVID-19 therapeutics found no association between treatment with RDV or CS and survival time despite current evidence supporting their use [13]. Conversely, this study emphasizes identifying the populations that are not likely to respond to treatments. Such knowledge is essential to prevent unnecessary complications from therapy use that might affect patient mortality, such as adverse drug effects, SBI as noted in our study group, and fungal infections, especially when critically ill. The study has limitations such as sample size, retrospective nature of the work, and uncertainty about the severity of the disease in the study group. However, the study highlights that machine learning can be a potential avenue to explore therapeutics in severe COVID-19 and help prevent complications from avoidable exposure to therapeutics.

In a meta-analysis of 11 randomized controlled trials, oseltamivir treatment reduced the risk of lower respiratory tract complications requiring antibiotic treatment by 28% overall and 37% among patients with confirmed influenza infection [14]. Animal experiments suggest that the influenza neuraminidase plays a role in the synergism between influenza virus infection and *Streptococcus pneumoniae*, thus providing a mechanism for Neuraminidase inhibitors' role in reducing the incidence of secondary bacterial pneumonia [15]. Perhaps, such studies with RDV will help understand its potential role in reducing bacterial infections. It is essential to know whether using antivirals in critically ill patients reduce the incidence of SBI that can be independently associated with increased mortality, hence supporting their use.

Excess antibiotic use in COVID-19 patients can threaten antimicrobial resistance and adherence to antimicrobial stewardship practices, further impacting mortality. As per one review, 72% of COVID-19 cases received antibacterial therapy though only 8% of the patients had bacterial/fungal co-infection [16]. Patients admitted with a critical illness are empirically placed on broad-spectrum antibiotics with the concern of MRSA (*Methicillin-Resistant Staphylococcus Aureus*) and drug-resistant gram-negative organisms like *Pseudomonas*. Data from a retrospective cohort study on the prevalence of MRSA in respiratory cultures of patients admitted with COVID-19 showed that intubated patients had more cultures obtained (78%) and that the prevalence of MRSA in respiratory cultures ranged from a low 0.6% on the day 3, to 5.7% on day 28, cumulatively [17]. Our study showed an overall MRSA and *Pseudomonas* prevalence of 5.15% on day 28. Our study results showed no significant difference in MRSA/*Pseudomonas* superinfections between the two adjusted cohorts (13.5% vs.13.5%, P = 1.00). Our findings support that continued empiric antibiotic usage for MRSA/*Pseudomonas* in patients with COVID-19 is likely not warranted. However, their use should be guided by local epidemiological data.

No significant associations with benefit were shown for hospital length of stay, mechanical ventilation use, clinical improvement, or clinical deterioration in a systematic review and meta-analysis of four peer-reviewed and published randomized clinical trials and six unpublished randomized clinical trials in patients with COVID-19 [18]. It is unknown if the convalescent plasma has mortality benefit for patients hospitalized with critical COVID-19 illness. To date, one published RCT in severe or life-threatening COVID-19, convalescent plasma therapy added to standard treatment, did not significantly improve the time to clinical improvement within 28 days and was halted early [19]. We observed a higher proportion of unmatched patients who received CS-RDV therapy also received convalescent plasma (20% vs. 3%, P<0.01) with no statistically significant difference between matched cohorts (14% vs 14%, P = 0.01). Administration of convalescent plasma in those who received was within one day of CS or CS-RDV therapy and was a part of initial management. Contrary to the propensity-matched single-center observational cohort study [20], our study results failed to show a mortality benefit independent of drug exposure with CS with or without RDV therapy. Lack of

knowledge about the protective titer concentrations can further complicate the studies mentioned. In a prospective, propensity score–matched study assessing the efficacy of COVID-19 convalescent plasma transfusion versus standard of care, transfusion of high anti-receptor binding domain (RBD) IgG titer COVID-19 convalescent plasma early in hospitalization was associated with a reduction in mortality in severe and/or critical COVID-19 patients [21]. Additional clinical trials are essential to further examine the efficacy of high titer plasma therapy.

We also analyzed for the potential confounding effect of interleukin-6 receptor antagonists. The patients who received them were small (10 in CS-RDV group and 23 in CS group) in number, precluding us from doing further data analysis.

## Limitations

The study's strengths are propensity score matching, the range of sensitivity analyses, falsification endpoints, and the data's real-world nature. Nevertheless, there are several limitations to this study. First, the level of detail possible with a manual medical record review may be missing with the use of an electronic medical record database. Second, despite rigorous statistical methods, there might be residual confounding that can impact the outcomes. Third, the sample size is small, impacting the power of the study. Fourth, the all-cause 28 day-hospital mortality rates reported in this study were estimated in patients requiring ventilatory support and hence did not reflect the mortality rate in all patients with COVID-19. Fifth, propensity score matching has its statistical issues, but our groups did not show a difference between unmatched and matched cohorts. Sixth, we do not have data on the exact date of the symptom of onset in these patients, and hence the efficacy of RDV in these patients may not be reflective of the available evidence.

## Conclusion

Treatment with CS-RDV therapy was non-inferior to CS monotherapy in critically ill patients in reducing mortality. However, combination therapy was associated with a significant decrease in the incidence of SBI in critically ill patients with no associated reduction in mortality rates. RDV use can be justified in those at high risk of infections if proven through further evidence. There is a dire need to explore new therapeutic options due to the scarcity of available therapeutic options and significant morbidity and mortality rates in critically ill patients. The current change in disease dynamics with evolving new genetic variants can complicate the disease trends, thus threatening scientific progress.

## Supporting information

**S1 Text.**
(DOCX)

## Acknowledgments

We thank Dr. Emmett for her immense support and for reviewing the manuscript. We thank Wiley editing services for their assistance in technical editing to meet the respective journal formatting requirements.

## Author Contributions

**Conceptualization:** Subhadra Mandadi.

**Data curation:** Frank Annie.

**Formal analysis:** Frank Annie.

**Methodology:** Subhadra Mandadi, Harish Pulluru, Frank Annie.

**Resources:** Frank Annie.

**Software:** Frank Annie.

**Supervision:** Subhadra Mandadi.

**Validation:** Subhadra Mandadi, Harish Pulluru, Frank Annie.

**Writing – original draft:** Subhadra Mandadi, Harish Pulluru, Frank Annie.

**Writing – review & editing:** Subhadra Mandadi, Harish Pulluru.

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
