## [Decision Letter · Decision Letter 0]

27 May 2021

PONE-D-21-08678

Comparative outcomes of combined corticosteroid and remdesivir therapy with corticosteroid monotherapy in ventilated COVID-19 patients

PLOS ONE

Dear Dr. Mandadi,

Thank you for submitting your manuscript to PLOS ONE. After careful consideration, we feel that it has merit but does not fully meet PLOS ONE’s publication criteria as it currently stands. Therefore, we invite you to submit a revised version of the manuscript that addresses the points raised during the review process.

We look forward to receiving your revised manuscript.

Kind regards,

Girish Chandra Bhatt, MD, FASN

Academic Editor

PLOS ONE

Journal Requirements:

2. In your ethics statement in the Methods section and in the online submission form, please provide additional information about the data used in your retrospective study. Specifically, please ensure that you have discussed whether all data were fully anonymized before you accessed them and/or whether the IRB or ethics committee waived the requirement for informed consent. If patients provided informed written consent to have data from their medical records used in research, please include this information.

Reviewers' comments:

Reviewer's Responses to Questions

**Comments to the Author**

1. Is the manuscript technically sound, and do the data support the conclusions?

Reviewer #1: Partly

Reviewer #2: Yes

2. Has the statistical analysis been performed appropriately and rigorously? 

Reviewer #1: No

Reviewer #2: Yes

3. Have the authors made all data underlying the findings in their manuscript fully available?

Reviewer #1: No

Reviewer #2: Yes

4. Is the manuscript presented in an intelligible fashion and written in standard English?

Reviewer #1: Yes

Reviewer #2: Yes

5. Review Comments to the Author

Reviewer #1: Great effort to answer a very pertinent question.

However the following points may kindly be looked at before presenting the document to the scientific community.

1. No ethics statement is mentioned. Was it attained?

2. It will be interesting to note, whether the authors tried to find out prior estimation of a sample size with a reasonable power.

3. Base line parameters : does it mean they were attained before initiation of the drugs concerned or during the intervention period?

4. From Table-2, it is clear that the base line parameters are different for the 2 groups. So there is an inherent problem to conclude about the outcomes and more so to generalize the outcomes.

Reviewer #2: This study has important implication regarding use of combination therapy of Corticosteroids and Remdesivir. These results will be helpful to clinician in decision making while treating seriously ill COVID-19 patients. My comments are as follows:

1. Conclusion need minor modification:

a. “Associated with a significant decrease in the incidence of secondary bacterial infections in critically ill COVID-19 patients.” make this as separate statement along with addition of part that this was not associated with decresed motality

2. Please discus results of your study in connection with this study

a. Lam C, Siefkas A, Zelin NS, Barnes G, Dellinger RP, Vincent JL, Braden G, Burdick H, Hoffman J, Calvert J, Mao Q, Das R. Machine Learning as a Precision-Medicine Approach to Prescribing COVID-19 Pharmacotherapy with Remdesivir or Corticosteroids. Clin Ther. 2021 Mar 29:S0149-2918(21)00128-4. doi: 10.1016/j.clinthera.2021.03.016. Epub ahead of print. PMID: 33865643; PMCID: PMC8006198.

3. Author can compare SBI between different steroids regimen (dexamethasone v/s methylprednisolone v/s hydrocortisone)

4. Duration of steroids in CS-RDV v/s CS

5. Whether convalescent plasma was used in refractory patients with CS-RDV

6. Any fungal infections in SBI especially mucormycosis

7. For combination therapy steroids were started on day 1 of RDV. It is not clear whether patients receiving RDV before initiation of steroids were included in the study.

6. PLOS authors have the option to publish the peer review history of their article (what does this mean?). If published, this will include your full peer review and any attached files.

Reviewer #1: **Yes: **Dr. Sagar Khadanga

Reviewer #2: **Yes: **Rakesh Kumar Pilania

---

## [Author Response · Author response to Decision Letter 0]

8 Jul 2021

Per journal requirements, we ensured that our manuscript meets PLOS ONE's style requirements, including those for file naming, and we utilized appropriate templates provided. We acknowledge our error in not providing a relevant ethics statement and data availability statement. We added the required to the methods section and submission system. Please see below for a point-by-point response to the Editors and reviewers’ comments and concerns. 

Editor comments: 

We apologize for not providing the necessary. Our study was approved under exemption by the CAMC (Charleston Area Medical Center) research and Grant's administration Institutional Review Board (study number 21-723) and received a waiver of informed consent on 1/11/21. 

The study used data from TriNetX, a global federated health research network that provided an anonymized dataset of electronic medical records (EMRs). Using the TriNetX network, a de-identified dataset of COVID-19 patients with a PCR confirmed SARS-COV-2 diagnosis, admitted to the Intensive Care Unit (ICU) aged 18 to 90, was identified in EMRs between January 20th, 2020, and February 9th, 2020. TriNetX is compliant with the Health Insurance Portability and Accountability Act (HIPAA), the US federal law that protects healthcare data privacy and security. TriNetX is certified to the ISO 27001:2013 standard and maintains an Information Security Management System (ISMS) to ensure the protection of the healthcare data it has access to and meet the HIPAA Security Rule requirements. Any data displayed on the TriNetX Platform in aggregate form, or any patient level data provided in a data set generated by the TriNetX Platform, only contains de-identified data as per the de-identification standard defined in Section §164.514(a) of the HIPAA Privacy Rule. The process by which the data is de-identified is attested to through a formal determination by a qualified expert as defined in Section §164.514(b)(1) of the HIPAA Privacy Rule. This formal determination by a qualified expert, refreshed in December 2020, supersedes the need for TriNetX’s previous waiver from the Western Institutional Review Board (IRB). The TriNetX network contains data provided by participating Healthcare Organizations (HCOs), each of which represents and warrants that it has all necessary rights, consents, approvals, and authority to provide the data to TriNetX under a Business Associate Agreement (BAA), so long as their name remains anonymous as a data source and their data are utilized for research purposes. The data shared through the TriNetX Platform are attenuated to ensure that they do not include sufficient information to facilitate the determination of which HCO contributed specific information about a patient. Further details about TriNetX processes and standardization of data are provided in S1 Text. For this study, we accessed the data from health care organizations in the United States.

The legal and ethical restrictions under which the data were provided do not allow for the data to be made publicly available. The data we used for this paper were acquired from TriNetX (https://www.trinetx.com/). Release and/or sharing of these data are not covered under our data use agreement with TriNetX. To gain access to the data, a request can be made to TriNetX (moc.xtenirt@nioj), but costs may be incurred, and a data sharing agreement would be necessary.

Reviewer #1 comments: Great effort to answer a very pertinent question.

However the following points may kindly be looked at before presenting the document to the scientific community.

1. No ethics statement is mentioned. Was it attained?

We apologize for not including the ethics statement. The ethics statement was obtained on 1/11/21. We added the details of the ethics statement to the online submission form and the methods section of the manuscript. The ethics statement under methods section on page 3 to page 4 reads as follows: 

"Our study was approved under exemption by the CAMC (Charleston Area Medical Center) research and Grant's administration Institutional Review Board (study number 21-723) and received a waiver of informed consent. The study used data from TriNetX, a global federated health research network that provided an anonymized dataset of electronic medical records (EMRs). Using the TriNetX network, a de-identified dataset of COVID-19 patients with a PCR confirmed SARS-COV-2 diagnosis, admitted to the Intensive Care Unit (ICU) aged 18 to 90, was identified in EMRs between January 20th, 2020, and February 9th, 2020. TriNetX is compliant with the Health Insurance Portability and Accountability Act (HIPAA), the US federal law that protects healthcare data privacy and security. TriNetX is certified to the ISO 27001:2013 standard and maintains an Information Security Management System (ISMS) to ensure the protection of the healthcare data it has access to and meet the HIPAA Security Rule requirements. Any data displayed on the TriNetX Platform in aggregate form, or any patient level data provided in a data set generated by the TriNetX Platform, only contains de-identified data as per the de-identification standard defined in Section §164.514(a) of the HIPAA Privacy Rule. The process by which the data is de-identified is attested to through a formal determination by a qualified expert as defined in Section §164.514(b)(1) of the HIPAA Privacy Rule. This formal determination by a qualified expert, refreshed in December 2020, supersedes the need for TriNetX’s previous waiver from the Western Institutional Review Board (IRB). The TriNetX network contains data provided by participating Healthcare Organizations (HCOs), each of which represents and warrants that it has all necessary rights, consents, approvals, and authority to provide the data to TriNetX under a Business Associate Agreement (BAA), so long as their name remains anonymous as a data source and their data are utilized for research purposes. The data shared through the TriNetX Platform are attenuated to ensure that they do not include sufficient information to facilitate the determination of which HCO contributed specific information about a patient. Further details about TriNetX processes and standardization of data are provided in S1 Text. For this study, we accessed the data from health care organizations in the United States". We added S1 Text file to the revised manuscript. 

2. It will be interesting to note, whether the authors tried to find out prior estimation of a sample size with a reasonable power.

We appreciate you for pointing this out. We did not do the prior estimation of sample size, and we acknowledge the relevance and significance. However, we performed a post hoc sensitivity analysis. Based on a 2-sided test with 80% power and a P value of 0.05 and with the mortality rates obtained, it was determined that a minimum of 140 (70 in each arm) would be required to detect the difference between the two groups. We added the above relevant information to the revised manuscript in the section of the outcome measures, page 12, lines 193-195. 

3. Base line parameters: does it mean they were attained before initiation of the drugs concerned or during the intervention period?

Thank you for the question. The retrospective cohort study included all patients with COVID-19 recorded in their EMRs from participating healthcare organizations through TriNetX. Patients with COVID-19 were identified following criteria provided by TriNetX using one or more of the following International Classification of Diseases, Tenth Revision, Clinical Modification (ICD-10-CM) codes in their EMRs, also mentioned in the manuscript Table 1. This was followed by identifying intubated patients admitted to ICU from the Emergency department who then sequentially received steroids and Remdesivir on day 1 of the hospital visit. We obtained comorbidities from these patients that can act as confounding factors to the measured outcome of mortality rates based on knowledge from the currently available evidence. We then used Propensity Score Matching of the two cohorts to control for confounding. 

4. From Table-2, it is clear that the base line parameters are different for the 2 groups. So there is an inherent problem to conclude about the outcomes and more so to generalize the outcomes.

We apologize for our failure in expressing our findings with clarity. There were differences in the baseline parameters among the unmatched cohorts for Age, Asian race, Hypertension, Diabetes, Congestive heart failure, Chronic kidney disease, Coronary artery disease, Chronic obstructive pulmonary disease, Chronic liver disease. To alleviate the possible confounding, we used Propensity Score Matching platform and analyzed the matched cohorts. Once the two cohorts were matched on these baseline parameters, we cannot study the effect of these characteristics on outcome measure. We thus established no difference in all-cause mortality between the matched study arms. 

Nevertheless, we explored further to test the potential differences between unmatched cohorts. We conducted an additional sensitivity analysis with Diabetes and Chronic kidney disease independently and found no statistical measure influencing mortality. The results indirectly inform us that the differences in baseline parameters may not serve as independent predictors of mortality once the patient needs ventilatory support but cannot prove the association due to limitations associated with our study design and sample size. We also quoted an article on autopsy findings published in Nature journal to support our findings. 

We added the following statement to the discussion section on page 11, continued to page 12, lines 233-241: " Among the unmatched cohorts, there were significant differences in preexisting health conditions. An independent additional sensitivity analysis with Diabetes and Chronic kidney disease showed no statistical measure influencing mortality rates independent of drug exposure. The results indirectly inform that the associated comorbidities might not have predictive effect on mortality rates in critically ill COVID-19 patients. An article published on autopsy findings of the 26 cases of hospitalized patients with COVID-19 evaluated the contribution of the preexisting health conditions to the risk of death. The investigators suggested that most patients whose median age was 70 years have died of COVID-19 illness with only contributory implications of preexisting health conditions to the mechanism of death [10]”.

We have added figures to address the valuable comment for the reviewer purpose only to the Response to Reviewers file. We did not add them to the revised manuscript. 

Reviewer #2 comments: This study has important implication regarding use of combination therapy of Corticosteroids and Remdesivir. These results will be helpful to clinician in decision making while treating seriously ill COVID-19 patients. My comments are as follows:

1. Conclusion need minor modification:

a. “Associated with a significant decrease in the incidence of secondary bacterial infections in critically ill COVID-19 patients.” make this as separate statement along with addition of part that this was not associated with decreased mortality. 

The observation is precise. We amended the conclusion section as per your suggestion. The new conclusion section on page 18 reads as: “Treatment with CS-RDV therapy was non-inferior to CS monotherapy in critically ill patients in reducing mortality. However, combination therapy was associated with a significant decrease in the incidence of SBI in critically ill patients with no associated reduction in mortality rates. RDV use can be justified in those at high risk of infections if proven through further evidence. There is a dire need to explore new therapeutic options due to the scarcity of available therapeutic options and significant morbidity and mortality rates in critically ill patients. The current change in disease dynamics with evolving new genetic variants can complicate the disease trends, thus threatening scientific progress”. 

We took this opportunity to incorporate Figure 3 that shows no association between the SBI and mortality rates among matched cohorts. We added Figure 3 in the Response to Reviewers file. 

2. Please discus results of your study in connection with this study

a. Lam C, Siefkas A, Zelin NS, Barnes G, Dellinger RP, Vincent JL, Braden G, Burdick H, Hoffman J, Calvert J, Mao Q, Das R. Machine Learning as a Precision-Medicine Approach to Prescribing COVID-19 Pharmacotherapy with Remdesivir or Corticosteroids. Clin Ther. 2021 Mar 29:S0149-2918(21)00128-4. doi: 10.1016/j.clinthera.2021.03.016. Epub ahead of print. PMID: 33865643; PMCID: PMC8006198.

We thank the reviewer for enlightening the importance of the above-mentioned study. We revised the manuscript by adding its pertinence in relation to our study results in the discussion section, page 15, lines 260-270 of the revised manuscript. The added portion reads as: “A recent study on machine learning as a precision medicine approach to identify a group of general inpatient COVID-19 patients who might benefit from COVID-19 therapeutics found no association between treatment with RDV or CS and survival time despite current evidence supporting their use [13]. Conversely, this study emphasizes identifying the populations that are not likely to respond to treatments. Such knowledge is essential to prevent unnecessary complications from therapy use that might affect patient mortality, such as adverse drug effects, secondary bacterial infections as noted in our study group, and fungal infections, especially when critically ill. The study has limitations such as sample size, retrospective nature of the work, and uncertainty about the severity of the disease in the study group. However, the study highlights that machine learning can be a potential avenue to explore therapeutics in severe COVID-19 and help prevent complications from avoidable exposure to therapeutics.”

3. Author can compare SBI between different steroids regimen (dexamethasone v/s methylprednisolone v/s hydrocortisone)

We agree that further elaboration on the incidence of SBI with various steroid regimens would be helpful. However, we could not perform the respective analysis as some patient cohorts received at least two different steroid types during the study period. Therefore, the results may not be representative of a single steroid regimen. We do recognize the limitation, and we added the following sentence on page 14, under the discussion section, lines 256-260 in the revised manuscript, which reads as: “Nevertheless, SBI was not shown to be an independent risk factor of mortality in our study. Most of the CS therapy cohort received a combination of steroids, and therefore, we could not determine the association of SBI with an independent steroid regimen.”

4. Duration of steroids in CS-RDV v/s CS

We appreciate the reviewer’s insightful suggestion. The study's primary aim was to compare the mortality rates between CS and combination therapy with CS and RDV at 48 hours and five days of therapy. RDV use is limited to five days in most US healthcare organizations, and steroid use is associated with mortality benefit when used for ten days. Therefore, we considered 48hrs and five days of therapy to match the cohorts to consider combination therapy and analyzed the respective mortality rates. Unfortunately, we could not run further analysis independently on the duration of steroid therapy and it was beyond the scope of this paper. 

5. Whether convalescent plasma was used in refractory patients with CS-RDV

We apologize for the lack of clarity. Administration of convalescent plasma in those who received was within one day of CS or CS-RDV therapy. Based on the available information, convalescent plasma was likely a part of initial management but not administered for refractory cases. We added the pertinent information to the revised manuscript for clarity on page 16 continued to page16 and 17, lines 304-305 of the discussion section, which is as follows, “Administration of convalescent plasma in those who received was within one day of CS or CS-RDV therapy and was a part of initial management.”

6. Any fungal infections in SBI especially mucormycosis

Thank you for a valuable and timely question. We ran the analysis using Mycoses diagnostic codes (B35-B49 ICD-10 codes), and we did not find any patients with fungal infections in those who had SBI and hence we did not add this information in the revised manuscript.

7. For combination therapy steroids were started on day 1 of RDV. It is not clear whether patients receiving RDV before initiation of steroids were included in the study.

Thank you for the question. The retrospective cohort study included all patients with COVID-19 recorded in their EMRs from participating healthcare organizations through TriNetX. Patients with COVID-19 were identified following criteria provided by TriNetX using one or more of the following International Classification of Diseases, Ninth Revision, and Tenth Revision, Clinical Modification (ICD-10-CM) codes in their EMRs, also mentioned in the manuscript Table 1. This is followed by identifying intubated patients admitted to ICU from the Emergency department who then sequentially received steroids and Remdesivir. We modified the manuscript for clarity based on your suggestion under the study population section on page 5, lines 115-116, which is as follows: “For the combination therapy study arm, patients sequentially received CS therapy, followed by RDV on day 1 of admission.”

---

## [Decision Letter · Decision Letter 1]

19 Jan 2022

PONE-D-21-08678R1Comparative outcomes of combined corticosteroid and remdesivir therapy with corticosteroid monotherapy in ventilated COVID-19 patientsPLOS ONE

Dear Dr. Mandadi,

Thank you for submitting your manuscript to PLOS ONE. After careful consideration, we feel that it has merit but does not fully meet PLOS ONE’s publication criteria as it currently stands. Therefore, we invite you to submit a revised version of the manuscript that addresses the points raised during the review process.

We look forward to receiving your revised manuscript.

Kind regards,

Girish Chandra Bhatt, MD, FASN

Academic Editor

PLOS ONE

Journal Requirements:

Reviewers' comments:

Reviewer's Responses to Questions

**Comments to the Author**

1. If the authors have adequately addressed your comments raised in a previous round of review and you feel that this manuscript is now acceptable for publication, you may indicate that here to bypass the “Comments to the Author” section, enter your conflict of interest statement in the “Confidential to Editor” section, and submit your "Accept" recommendation.

Reviewer #2: All comments have been addressed

2. Is the manuscript technically sound, and do the data support the conclusions?

Reviewer #2: Partly

3. Has the statistical analysis been performed appropriately and rigorously? 

Reviewer #2: Yes

4. Have the authors made all data underlying the findings in their manuscript fully available?

Reviewer #2: Yes

5. Is the manuscript presented in an intelligible fashion and written in standard English?

Reviewer #2: Yes

6. Review Comments to the Author

Reviewer #2: 1. Abstract: Sentence is staring with abbreviation. Please reveres that RDV (Remdesivir) to Remdesivir (RDV). Similarly need to be looked at other places as well.

2. All cause 28-day mortality rates between matched cohorts (28/28) calculated after five days of CSRDV and CS therapy were similar (51% vs. 70%; P=0.11). Although mortality is not showing statistical significance. However, difference is approximately 19%.

3. ‘Patients with elevated transaminases more than five times the standard upper normal limit was excluded from the CSRDV study arm’. This is confusion. Whether authors have prospectively enrolled the patients or analyzed from database.

4. Line 137 Secondary bacterial infection write as SBI

5. Authors have used both Covid-19 as well as COVID-19. To maintain uniformity one abbreviation should be used. Similarly, syntax and grammar need to be looked throughout the manuscript

7. PLOS authors have the option to publish the peer review history of their article (what does this mean?). If published, this will include your full peer review and any attached files.

Reviewer #2: **Yes: **Rakesh Kumar Pilania

---

## [Author Response · Author response to Decision Letter 1]

1 Feb 2022

Reviewer # 2 comments:

1. Abstract: Sentence is staring with abbreviation. Please reveres that RDV (Remdesivir) to Remdesivir (RDV). Similarly need to be looked at other places as well.

Thank you for the precise observation. These will be very helpful as reminders for our future work. We have made necessary corrections throughout the manuscript where necessary, including the abstract. 

 2. All cause 28-day mortality rates between matched cohorts (28/28) calculated after five days of CSRDV and CS therapy were similar (51% vs. 70%; P=0.11). Although mortality is not showing statistical significance. However, difference is approximately 19%.

We appreciate the reviewer reminding us to add necessary details to the manuscript, which are critical. We amended the sentence in the first paragraph of the discussion section as suggested. The new sentence added reads as: "The all-cause 28-day mortality rates between matched cohorts (28/28) calculated after five days of CS-RDV and CS therapy were 51% and 70%, respectively (P=0.11). The difference between the matched cohorts appears to be numerically significant at 19% but did not reach statistical significance, necessitating further evaluation with larger sample size."

 3. 'Patients with elevated transaminases more than five times the standard upper normal limit was excluded from the CSRDV study arm'. This is confusion. Whether authors have prospectively enrolled the patients or analyzed from database.

We appreciate the reviewer educating us on crafting the sentences carefully to avoid confusion. We amended sentences 109 -111 in the first paragraph of the study population section as per the suggestion. The new sentence reads as: "While selecting patients retrospectively for the CS-RDV group, we did not include patients with elevated transaminases more than five times the standard normal upper limit."

 4. Line 137 Secondary bacterial infection write as SBI

Thank you for the feedback. We have made necessary corrections throughout the manuscript where necessary. 

 5. Authors have used both Covid-19 as well as COVID-19. To maintain uniformity one abbreviation should be used. Similarly, syntax and grammar need to be looked throughout the manuscript

We apologize for our error. We have made necessary corrections throughout the manuscript where necessary.

---

## [Editor Report · Decision Letter 2]

9 Feb 2022

Comparative outcomes of combined corticosteroid and remdesivir therapy with corticosteroid monotherapy in ventilated COVID-19 patients

PONE-D-21-08678R2

Dear Dr. Mandadi,

We’re pleased to inform you that your manuscript has been judged scientifically suitable for publication and will be formally accepted for publication once it meets all outstanding technical requirements.

Kind regards,

Girish Chandra Bhatt, MD, FASN

Academic Editor

PLOS ONE
---

## [Editor Report · Acceptance letter]

14 Feb 2022

PONE-D-21-08678R2 

Comparative outcomes of combined corticosteroid and remdesivir therapy with corticosteroid monotherapy in ventilated COVID-19 patients 

Dear Dr. Mandadi:

I'm pleased to inform you that your manuscript has been deemed suitable for publication in PLOS ONE. Congratulations! Your manuscript is now with our production department. 

Kind regards, 

on behalf of

Dr. Girish Chandra Bhatt 

Academic Editor

PLOS ONE